# Effectiveness and safety of oral lactococci-based vaccine encoding triple common allergens to prevent airway allergy in mice

**Mey-Fann Lee[1], Chu-Hui Chiang[2], Shyh-Jye Lin[3], Chi-Sheng Wu[4], Yi-Hsing Chen[4,5]\***

**1** Department of Medical Research, Taichung Veterans General Hospital, Taichung, Taiwan, **2** Department of Plant Medicine, National Pingtung University of Science and Technology, Pingtung, Taiwan, **3** School of Medical Laboratory and Biotechnology, Chung Shan Medical University, Taichung, Taiwan, **4** Division of Allergy, Immunology and Rheumatology, Taichung Veterans General Hospital, Taichung, Taiwan, **5** School of Medicine, National Yang Ming Chiao Tung University, Taipei, Taiwan

\* ysanne@vghtc.gov.tw

**Data Availability Statement:** All relevant data are within the manuscript and its Supporting Information files.

## Abstract

Allergic airway disease is the most common chronic airway inflammatory disorder in developed countries. House dust mite, cockroach, and mold are the leading allergens in most tropical and subtropical countries, including Taiwan. As allergen avoidance is difficult for patients allergic to these perennial indoor allergens, allergen-specific immunotherapy (ASIT) is the only available allergen-specific and disease-modifying treatment. However, for patients sensitized to multiple allergens, ASIT using each corresponding allergen is cumbersome. In the present study, we developed a recombinant *L. lactis* vaccine against the three most common indoor aeroallergens and investigated its effectiveness for preventing respiratory allergy and safety in mice. Three recombinant clones of Der p 2 (mite), Per a 2 (roach), and Cla c 14 (mold) were constructed individually in pNZ8149 vector and then electroporated into host strain *L.lactis* NZ3900. BALB/c mice were fed with the triple vaccine 5 times per week for 4 weeks prior to sensitization. The effectiveness and safety profile were then determined. Oral administration of the triple vaccine significantly alleviated allergen-induced airway hyper-responsiveness in the vaccinated mice. The allergen-specific IgG2a was upregulated. IL-4 and IL-13 mRNA expressions as well as inflammatory cell infiltration in the lungs decreased significantly in the vaccinated groups. No body weight loss or abnormal findings in the liver and kidneys were found in any of the groups of mice. This is the first report to describe a triple-aeroallergen vaccine using a food-grade lactococcal expression system. We developed a convenient oral delivery system and intend to extend this research to develop a vaccination that can be self-administered at home by patients.

## Introduction

Allergic airway disease is the most common chronic IgE-mediated hypersensitivity in developed countries and global rates continue to rise [1–3]. In addition to outdoor air pollution and allergens, it has been reported that eight indoor agents are highly involved in the development

**Funding:** Funding Sources: This study was supported by grants from Taichung Veterans General Hospital (TCVGH-1077313C and 1087316C).

**Competing interests:** The authors have declared that no competing interests exist.

and exacerbation of asthma: cockroaches, dust mites, cat dander, dog dander, respiratory viruses, fungi, nitrogen dioxide, and environmental tobacco smoke [4]. Among them, aeroallergens derived from dust mite, cockroaches, and molds are the most common sensitizers and elicitors of respiratory allergy in tropical and subtropical regions in the world, including Taiwan [5–9].

Although allergen avoidance is theoretically the best way of preventing clinical manifestation of allergy, the pervasive contamination by some aeroallergens, such as dust mites, cockroaches and molds in the household environment, means that in practical terms exposure is inevitable. Allergen-specific immunotherapy (AIT) is the only disease-modifying approach with long-lasting effects through induction of allergen-specific blocking antibodies and regulatory T cells to achieve tolerance to the corresponding allergens [10, 11]. Conventional subcutaneous immunotherapy, though effective, requires frequently repeated injection of natural allergen extracts containing a wide variety of undesirable proteins, which has thus limited its applicability [12, 13].

It has been shown in animal models that oral feeding of protein antigens can downregulate systemic immune responses, known as oral tolerance [14, 15]. Oral administration of therapeutic molecules theoretically offers advantages such as ease of administration and reduction in adverse effects. However, aside from the recent approval of an oral peanut immunotherapy agent [16, 17], many of the oral immunotherapeutic agents for aeroallergens failed to demonstrate clinical effectiveness [18–20]. To date, a limited number of sublingual-pastille-like immunotherapeutic vaccines for aeroallergens have demonstrated clinical efficacy and safety and have been approved for clinical use [21–26].

In the past few decades, the DNA sequences of the most common allergens have been identified and the corresponding allergens can been produced as recombinant allergens [27, 28]. As a result of these advances, genetically recombinant allergen proteins may be used as source of allergen-specific immunotherapy to improve the quality and safety of allergy vaccines.

Gram-positive non-pathogenic lactic acid bacteria have long been widely used in the food industry. The protective or modulatory effects of recombinant *L.lactis* strains for several diseases have been verified in animal models and clinical trials [29–35]. In this study, we developed a recombinant *L. lactis* vaccine containing three of the most common indoor aeroallergens and investigated its effectiveness and safety for preventing respiratory allergy in mice.

## Materials and methods

### Bacterial strain and vector

The *L. lactis* NZ3900 strain and plasmid pNZ8149 used in this work were purchased from MoBiTec (Goettingen, Germany). NZ3900 was used for food-grade expression based on its ability to grow on lactose. Deletion of the *lacF* gene renders this strain unable to grow on lactose unless *LacF* is provided in a plasmid. pNZ8149 contains the *LacF* gene for food-grade selection for growth on lactose and a nisA promoter for gene expression by nisin induction. Nisin is a 34-amino acid anti-microbial peptide and is now widely permitted as a *food-safe* preservative.

### Construction of pNZ8149-Per a 2/Der p 2/Cla c 14 and transformation by electroporation

To construct the recombinant plasmid expressing the fusion genes under the control of the regulative promoter nisA, three primer pairs for each of the allergen genes were used for the

**Table 1. Specific primers of allergens used in the study.**

| Primers | Sequences | Restriction site | cDNA /MW |
|---|---|---|---|
| Per a 2-F | CCATGGATCCAGTCGTCGTTCCT | *Nco*I | 995 bp/38 kDa |
| Per a 2-R | TCTAGACTACAGTTCTTCTACGGA | *Xba*I | |
| Der p 2-F | CTGCAGCTGATCAAGTTGATGTTAAAGAT | *Pst*I | 404 bp/16 kDa |
| Der p 2-R | TCTAGATTAATCACGAATTTTAGCATG | *Xba*I | |
| Cla c 14-F | CTGCAGTGTCTTCCTCCCTCGACCA | *Pst*I | 971 bp/36 kDa |
| Cla c 14-R | TCTAGACTTCTCGATCTTCTCGCGGA | *Xba*I | |

The underlined nucleotides indicate the added restriction sites for cloning.

polymerase chain reaction, as listed in Table 1. The amplified sizes of cDNA and the molecular weights of the derived proteins are shown in Table 1. The amplified PCR products were cloned into the pCR2.1 vector and confirmed by DNA sequencing with an automated DNA analyzer (ABI Prism 3700). Then, the three fragments of Per a 2, Der p 2, and Cla c 14 were subcloned into the *Nco*I/*Xba*I or *Pst*I/*Xba*I sites of the pNZ8149 vector in-frame. The constructed plasmids were extracted, purified, and transformed into *L.lactis*. Briefly, NZ3900 cells were cultured in an M17 broth (OXOID, UK) with 0.5 M sucrose, 2.5% glycine, and 0.5% glucose (M17B-S/G/Glucose) at 30˚C until the $OD_{600}$ of 0.2~0.3 was reached; it was then washed and resuspended in 1/100 volume of 0.5 M sucrose containing 10% glycerol. Competent cells were added to the ligation mixture and electroporated using a Gene Pulser (2500 V, 200 Ω, 25 μF, 5 ms, Bio-Rad, USA). The electroporated mixture was immediately diluted in 1 ml of M17B-S/G/Lactose containing 20 mM $MgCl_2$ and 2 mM $CaCl_2$ at 30˚C for 1 hour. Then the mixture was plated onto Elliker plates according to the manufacturer's instructions (MoBiTec, Germany). The lactose-positive colonies were visible in yellow color after 48 hours of incubation at 30˚C.

## Protein expression with the food-grade inducer nisin

In this study, the recombinant allergens Per a 2, Der p 2, and Cla c 14 were intracellularly produced under nisin induction in *L. lactis* NZ3900 clones. The selected *L.lactis* recombinant clones were propagated in M17 medium containing 0.5% lactose as the sole carbon source at 30˚C. The *L. lactis* strains harboring plasmids with Per a 2/Der p 2/Cla c14 genes were grown until an $OD_{600}$ of 0.1~0.2 was reached and induced with different concentrations of nisin (0~400 ng/ml, Sigma, Missouri, USA) for 1~16 hours. The harvested cells were monitored for protein expression by SDS-PAGE and immunodetection using rabbit anti-rPer a 2/Der p 2/Cla c 14 polyclonal antibodies.

## SDS-PAGE and western blot

Harvested cells of *L.lactis* were suspended in PBS and disrupted by sonication using a Branson digital sonifier for 30 minutes on ice. After centrifugation at 8000 rpm for 10 minutes, the supernatants were subjected to protein quantitation by the Bradford method using BSA as standards (Bio-Rad, Hercules, CA, USA). Cell lysates of *L.lactis* were loaded on a 4% polyacrylamide stacking gel with a 12% separating gel, and the gel was run with discontinuous buffer by Laemmli's method. After electrophoresis, gels were fixed and stained with 0.2% Coomassie brilliant blue R250. For immunoblotting, the gels were transferred to nitrocellulose membranes and blocked with 2% BSA. The membranes were then probed with rabbit anti-rPer a 2/rDer p 2/rCla c 14 antibodies (lab-prepared) followed by a peroxidase-labeled goat anti-rabbit

IgG antibody (10000-fold dilution, Millipore). Detection was performed using 3-amino-9-ethylcarbazole (AEC) as an enzyme substrate.

## Batch fermentation of recombinant L.lactis strains for oral vaccination

The recombinant *L.lactis* strains NZ3900 harboring pNZ8149-Per a 2/Der p 2/Cla c 14 were cultured in a bench-top fermentor (Firstek, Taiwan) equipped with a digital pH controller, a temperature-control system, a dissolved oxygen sensor, and a blender. Prior to fermentation, the *L. lactis* strain was propagated twice in M17 broth and statically incubated in a 30˚C incubator for 48 hours. An inoculum comprising 1% (v/v) of the seed culture was transferred to a 5-L fermentor containing 3-L M17 medium for culture. The fermentor was operated at 30˚C and the pH was controlled at 6.8~7.2 by automatically adding 2N NaOH solution with a pump. A slow agitation (40 rpm) was maintained to keep the broth homogeneous. Samples were withdrawn aseptically from the broth for analysis of the optical density at 600 nm at regular time intervals. Recombinant protein inducer nisin was added to the broth with optimal dosage and time determined by pre-experiments. The harvested cell pellets were washed with PBS and then resuspended in PBS to a concentration of $1 \times 10^{11}$ CFU/ml. The expressed rPer a 2/rDer p 2/rCla c 14 protein was determined by immunoblotting using purified *E.coli*-derived rPer a 2 /Der p 2/ Cla c 14 as standards, respectively. The corresponding control strain NZ3900 harboring empty plasmid was treated in the same way.

## Preparation of Per a 2/Der p 2/Cla c 14 recombinant proteins for sensitization

*E.coli*-expressed Per a 2/Der p 2/Cla c 14 recombinant proteins (*E*-rPer a 2/E-rDer p 2/*E*-rCla c 14) were purified by rapid affinity column chromatography (Novagen, Darmstadt, Germany). The recombinant proteins were further purified by Endotoxin Detoxi-Gel (Pierce, Illinois, USA) and sterilized with a 0.22-μm syringe filter (Millipore, Billerica, MA, USA). Finally, the protein concentration was determined by the Coomassie brilliant G-250 protein-dye binding method of Bradford with bovine serum albumin as a standard according to the manufacturer's instructions (Bio-Rad, Hercules, CA, USA).

## Animals

BALB/c mice (6 weeks old, female) were purchased from the National Laboratory Animal Center, Taiwan, and raised under specific pathogen-free conditions. All animal experiments were reviewed and approved by the Institutional Animal Care and Use Committee of Taichung Veterans General Hospital (no. La1071603).

## Experimental design of prophylactic *L.lactis* vaccination

The scheme of the experimental design is shown in Fig 1. The mouse model was established as previously described [29]. To evaluate the prophylactic effect of recombinant *L.lactis* vaccination, BALB/c mice were intragastrically (IG) administered 200 μl of recombinant *L.lactis* in PBS containing 1 μg each of the rPer a 2, rDer p 2 and rCla c 14 once a day on weekdays for 4 weeks before sensitization. Mice were fasted for 2 hours before each oral treatment. On days 21, 28, and 35, mice were sensitized intraperitoneally (IP) with 2 μg each of *E*-rPer a 2/E-rDer p 2/E-rCla c 14/1 mg alum. Between day 49 and day 56, the mice were challenged intratracheally (IT) with *E*-rPer a 2/*E*-rDer p 2/*E*-rCla c 14 for four consecutive days. To check the response of sensitization, airway hyperresponsiveness (AHR) was performed on day 56 after the fourth IT challenge. Serum samples were collected from the retro-orbital venous plexus bi-

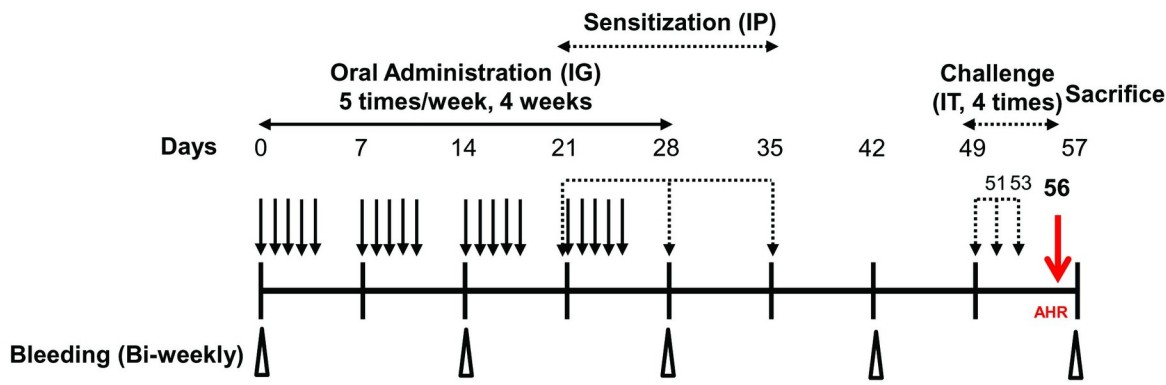

**Fig 1. Experimental design of prophylactic triple allergy vaccine in a murine model.** BALB/c mice were intragastrically (IG) administered triple vaccine or wild-type *L.lactis* once a day on weekday for 4 weeks. On days 21–35, mice were sensitized intra-peritoneally (IP) once a week with three shots of the mixture of Per a 2/Der p 2/Cla c 14 or PBS as the negative control. From days 49–56, all groups were challenged intratracheally (IT) for four consecutive days and sacrificed on day 57.

weekly and stored at -20°C until analysis. All mice were sacrificed on day 57 and spleen, lung, kidney, and liver were removed for further study.

## Measurement of specific antibodies by ELISA

Serum-specific IgE, IgG1 and IgG2a antibodies were determined by ELISA with the required antibodies purchased from BD Pharmingen (San Jose, CA, USA). Microtiter plates were coated with specific antigens for 2 hours at 37°C. After washing with PBST, plates were blocked with 2% BSA for 2 hours at room temperature. Murine sera were diluted (1:10 for IgE or 1:100 for IgGs) in PBST and incubated at room temperature for 2 hours. For IgE measurement, the plates were incubated with biotin-conjugated rat anti-mouse IgE (1:1000) for 2 hours at room temperature. Subsequently, horseradish peroxidase-conjugated streptavidin (1:4000) (Sigma) was added to the plates for 1 hour followed by addition of tetramethylbenzidine (TMB) as a substrate for 10 minutes. Finally, the reaction was stopped by adding 2M sulfuric acid. For IgG measurement, the plates were incubated with horseradish peroxidase-conjugated rabbit anti-mouse IgG1 or IgG2a (1:10000) for 2 hours at room temperature and developed by adding 2,2'-azino-bis(3-ethylbenzthiazoline-sulfonic acid (ABTS, Sigma). Then, the optical density was analyzed on a Sunrise Absorbance Reader (TECAN, Austria) at 450 or 415 nm, respectively.

## Measurement of airway hyperresponsiveness (AHR)

AHR, an exaggerated bronchoconstrictor response to inhaled stimuli, is a key feature of asthma, which is closely related to the severity and frequency of symptoms. The airway resistance of mice was measured using a whole-body Buxco mouse plethysmograph (Buxco, NY,

USA) after the last challenge. Mice were placed in the main chamber and challenged with aerosolized methacholine at concentrations of 50 mg/ml generated by a nebulizer (Buxco aerosol distribution system). Methacholine is an inhaled drug that causes narrowing of the airways in the lungs. The methacholine challenge test is a type of bronchial challenge test used to help diagnose asthma. The degree of bronchoconstriction was measured and averaged for 3 minutes after each nebulization. Data are expressed as enhanced pause (Penh) by the following equation: Penh = pause × (PEP/PIP). Pause, PEP, and PIP represent the expiration time, the peak expiratory pressure, and the peak respiratory pressure, respectively.

## Organ histopathology

Histopathology of vital organs was carried out to rule out any vaccine-induced organ toxicity. At the end of the experiments, the lungs, livers, and kidney were removed and processed for routine histological analysis. Briefly, tissue was fixed with 10% formalin and embedded in paraffin. Four-micrometer sections were cut and stained with a hematoxylin and eosin (H&E) staining kit (CIS-Biotechnology, Taiwan) and corresponding images were captured using an Olympus BX51 microscopic/DP71 Digital Camera System (Nagano, Japan). Moreover, the infiltrating inflammatory cells of lung sections were quantified by light microscopy under 400-fold view.

## Real-time PCR

To get a better idea of how various molecules were expressed in the lungs of the experimental mice, we used quantitative PCR on a StepOnePlus$^{TM}$ system (Applied Biosystems, CA, USA) to measure the expression of cytokines. The predesigned primer sequences are listed in Table 2. In brief, cDNA was prepared from 1 μg total RNA using a SuperScript III kit (Invitrogen, Carlsbad, CA). A total volume of 10 μl of PCR mixture, which included 5 μl of Real-Time SYBR Green/ROX PCR master mix from Applied Biosystems (Life technologies, CA, USA), 4 μl of double-distilled H$_2$O, and 1 μl of template cDNA, were added in each well of the PCR array. PCR amplification was conducted with an initial 10-min step at 95°C followed by 40 cycles of 95°C for 15 seconds and 60°C for 1 minute. The fluorescent signal from SYBR Green was detected immediately after the extension step of each cycle, and the cycle at which the product was first detectable was recorded as the cycle threshold. Gene expression levels were quantified relative to the expression of β-actin using the optimized comparative Ct ($2^{-\Delta\Delta Ct}$) value method.

**Table 2. The sequences of gene-specific primers for mice used in real-time PCR.**

| Gene (mouse) | | Sequences | Produce size (bp) |
|---|---|---|---|
| IL-4 | F | 5' AGC CAT ATC CAC GGA TGC GAC AAA 3' | 176 |
| | R | 5' AAT ATG CGA AGC ACC TTG GAA GCC 3' | |
| IL-13 | F | 5' AGA CCA GAC TCC CCT GTG CA 3' | 123 |
| | R | 5' TGG GTC CTG TAG ATG GCA TTG 3' | |
| IL-10 | F | 5' CCA AGC CTT ATC GGA AAT GA 3' | 155 |
| | R | 5' AGG GGA GAA ATC GAT GAC AG 3' | |
| GM-CSF | F | 5' GCT GCT GAG ATG AAT GAA AC 3' | 265 |
| | R | 5' AGT CAA AGG GGA TGA CAA G 3' | |
| β-actin | F | 5' GGC CAA CCG TGA AAA GAT GA 3' | 251 |
| | R | 5' CAC GCT CGG TCA GGA TCT TC 3' | |

### Statistical analysis

Statistical analysis was performed using IBM SPSS software version 22 (IBM Corporation, Armonk, NY, USA). All values are expressed as means ± SD, and differences between groups were analyzed by one-way analysis of variance with a Bonferroni multiple comparison test. P-values less than 0.05 were considered to be significant.

## Results

### Construction, expression, and immunoreactivity of the three recombinant *L.lactis*-Per a 2/Der p 2/Cla c 14

Three strains of recombinant *L.lactis* producing Per a 2, Der p 2, and Cla c 14 were constructed and the insertions into pNZ8149 plasmid were confirmed by restriction enzyme digestion as shown in Fig 2. The expression of Per a 2, Der p 2, and Cla c 14 via nisin induction in the

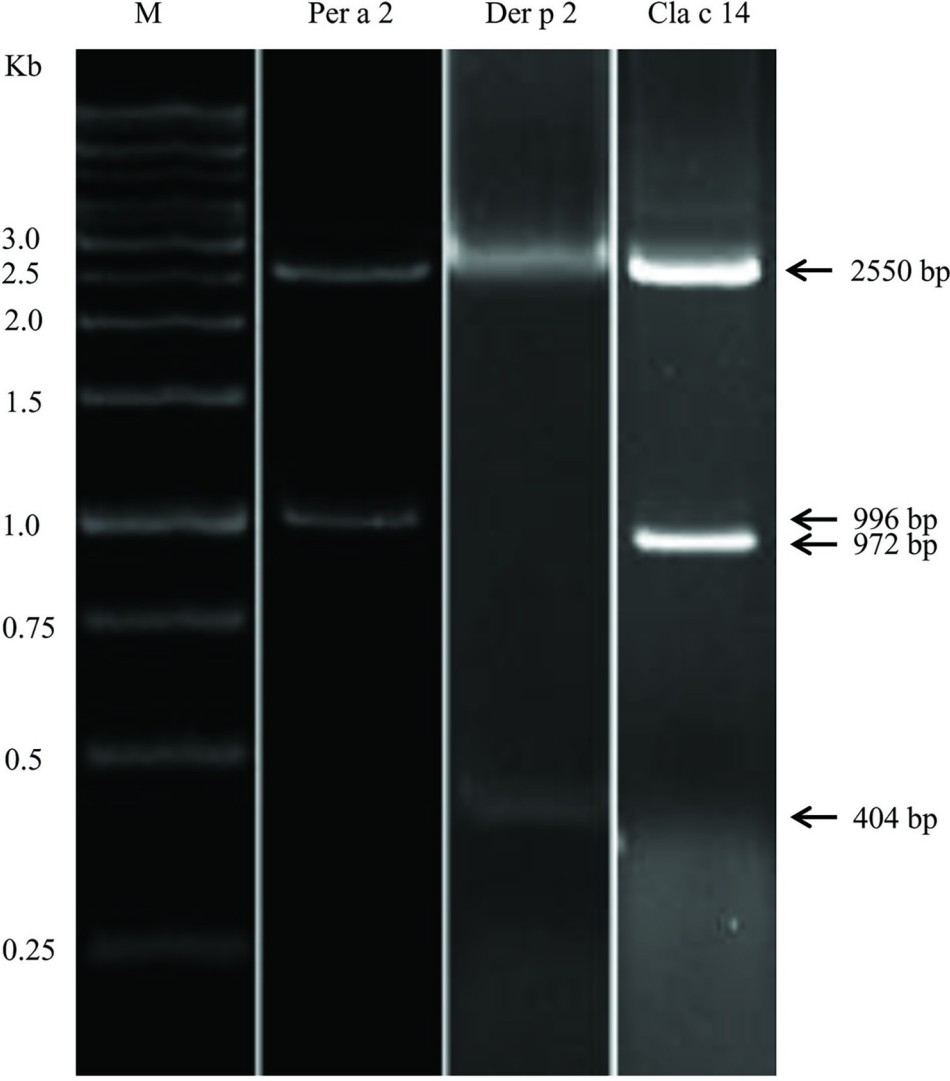

**Fig 2. Ethidine bromide-stained agarose gel of the digested recombinant plasmids as indicated.** The purified DNA of pNZ8149-Per a 2, pNZ8149-Der p 2 and pNZ8149-Cla c 14 were digested with restriction enzymes of *Nco*I/*Xba*I, *Pst*I/*Xba*I and *Pst*I/*Xba*I, respectively. Numbers at left indicate size of standard DNA fragments in kilo base pairs (Kb).

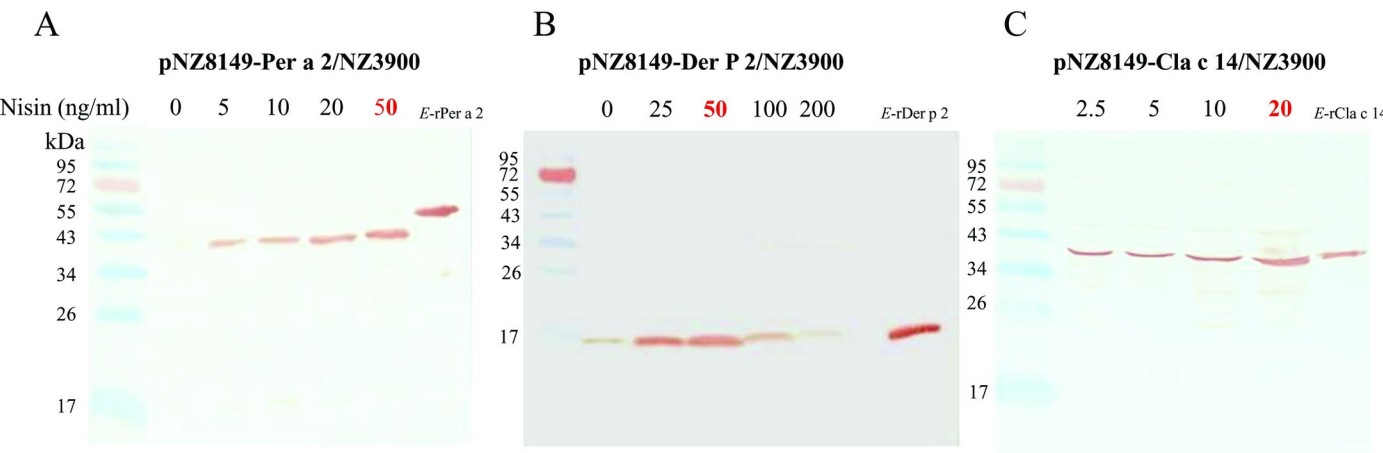

**Fig 3.** Immunoblot analyses of Per a 2 (A), Der p 2 (B) and Cla c 14-producing (C) engineered *L. lactis*. Intracellular production of the allergens were detected in total cell extracts after various doses of nisin induction using Per a 2, Der p 2 or Cla c 14-specific polyclonal antibodies, respectively. The results indicated that the selected recombinant *L.lactis* strain successfully expressed Per a 2, Der p 2 and Cla c 14 proteins.

recombinant *L.lactis* NZ3900 was evaluated by Western blots using in-house anti-Per a 2 (Fig 3A), anti-Der p 2 (Fig 3B), and anti-Cla c 14 (Fig 3C) polyclonal antibodies, respectively. Maximal recombinant protein productions were reached at an $OD_{600}$ of 0.16 under the following culture condition: 30°C using 50 ng/ml (for Per a 2 and Der P 2) or 20 ng/ml (for Cla c 14) nisin with 3 hours of induction.

## Recombinant *L. lactis* triple vaccine modulated allergen-specific antibody responses

The prophylactic potential of the triple allergy vaccine was assessed in a lab-developed Per a 2/ Der p 2/Cla c 14 sensitization murine model. Our model revealed that serum-specific IgE levels against corresponding allergens were significantly higher in the sensitized/not vaccinated and sensitized/vector only groups compared with the naive group (Fig 4A, 4C and 4E). However, the allergenic-specific IgE responses were significantly reduced in the sensitized/triple vaccinated group (Fig 4A, 4C and 4E) and the specific IgG2a levels increased (Fig 4B, 4D and 4F) in sera compared with the not vaccinated group or naive group, respectively.

## Recombinant *L. lactis* triple vaccine alleviated airway hyperresponsiveness and pulmonary inflammation

To explore the prophylactic potential of triple vaccine on the development of allergic asthma, mice underwent IP sensitization and IT challenge with the combination of three recombinant allergens. After the final challenge, mice were treated with 50 mg/ml of methacholine aerosol and Penh value was measured. In the not vaccinated and vector alone groups, mice showed markedly increased Penh upon methacholine exposure. In the triple vaccinated group, mice showed a significant reduction of Penh either compared with not vaccinated or VO groups (Fig 5).

Furthermore, the cytokine mRNA expressions in the lungs of mice in each group were determined by real-time PCR as shown in Fig 6. There was significant downregulation of mRNA levels of IL13 (Fig 6A), IL-4 (Fig 6B), IL10 (Fig 6C), and GM-CSF (Fig 6D) in the triple vaccine group compared with the not vaccinated group. Additionally, histopathologic features of the murine lungs are shown in Fig 7A. The examination of the lung tissues from either the

not vaccinated or vector alone group revealed numerous inflammatory cells surrounding the

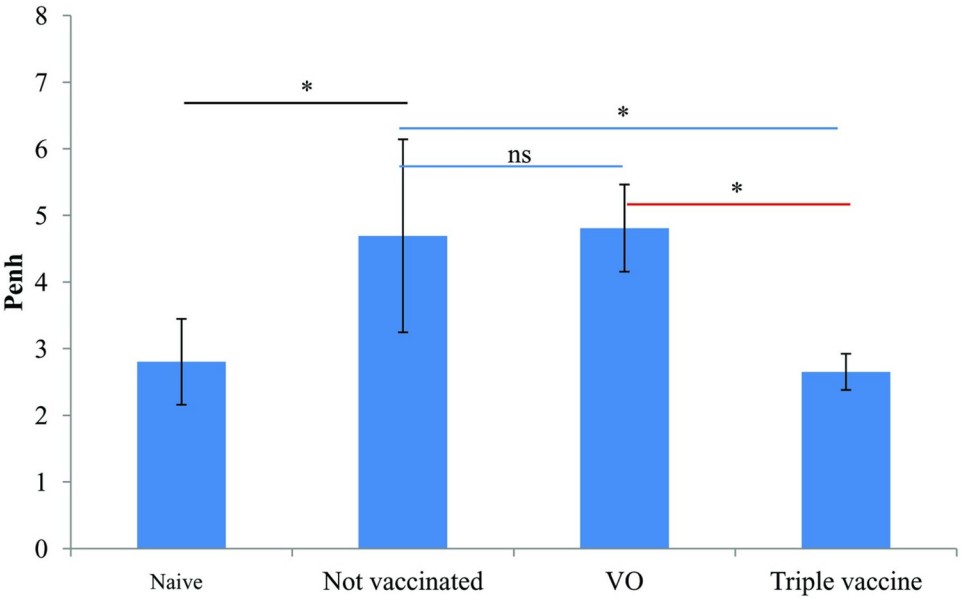

**Fig 4.** Serum levels of specific-IgE (A C, E) and -IgG2a (B, D, F) antibodies for individual antigens among four groups at week 8 determined by ELISA. Results are mean±SD of 6 mice from each group. $^*p < 0.05$; $^{**}p < 0.01$ by one-way analysis of variance with the Bonferroni multiple range test.

airways (Fig 7A), whereas the oral triple vaccine group produced a marked decrease in cellular infiltration and changes similar to the naïve group (Fig 7A). The results of differential cell counts revealed that oral recombinant *L.lactis* triple vaccine could significantly reduce infiltration of total inflammatory cells and eosinophils in lungs (Fig 7B).

**Fig 5. Effect of triple vaccine on airway hyperresponsiveness in mice sensitized to 3 allergens.** Mean enhanced pause (Penh) values were evaluated at 50 mg/ml of methacholine in the four groups. $^*p < 0.05$ by one-way analysis of variance with the Bonferroni multiple range test.

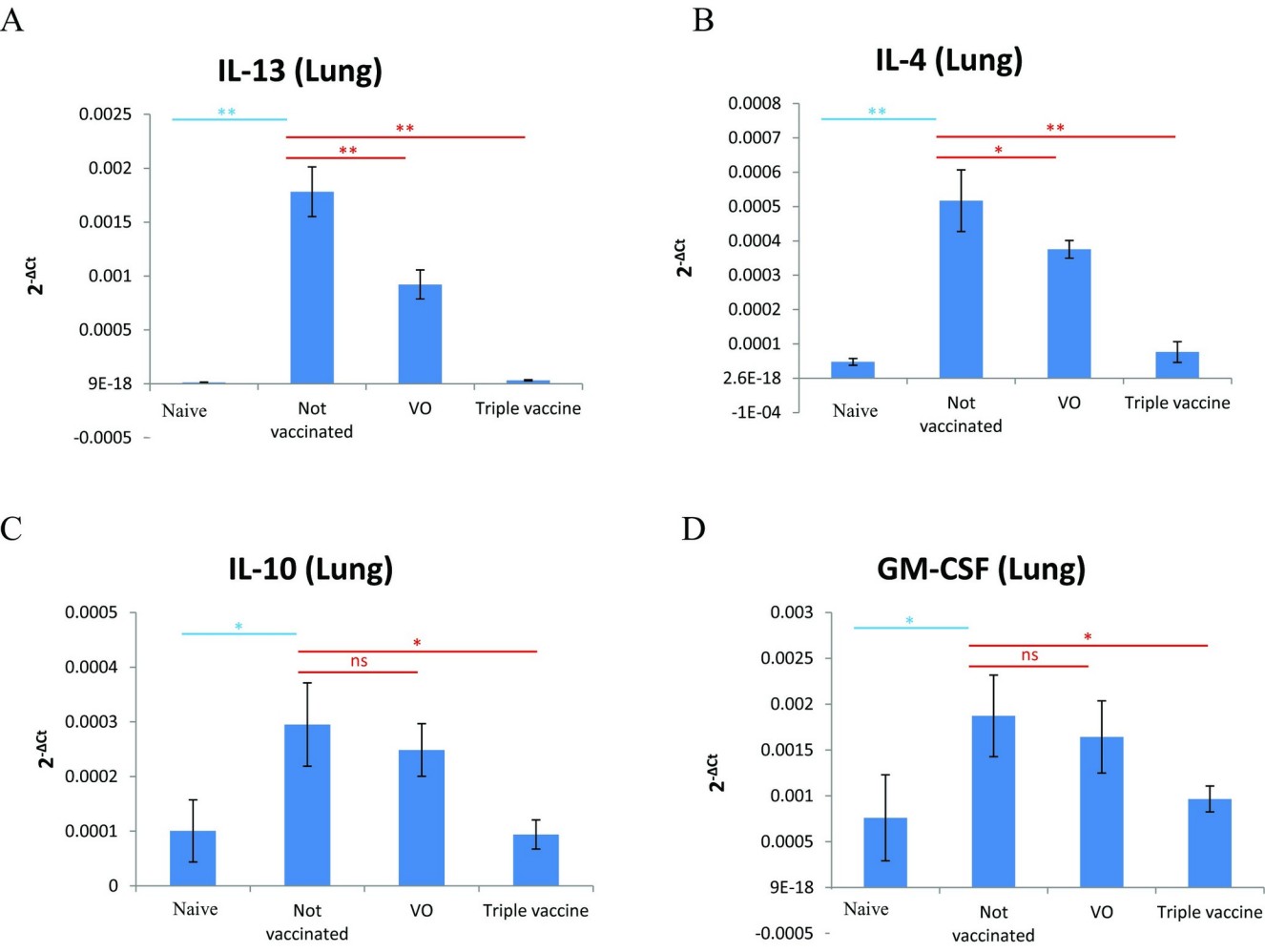

**Fig 6. Cytokine mRNA levels of lungs from each group of mice by real-time PCR.** (A) IL-13, (B) IL-4, (C) IL-10, (D) GM-CSF. Data are expressed as mean of $2^{-\Delta Ct}$ ±SD. The statistical significance of differences among groups was assessed by the Bonferroni multiple range test. *denoted $p<0.05$; **denoted $p<0.01$.

### Safety testing in the vaccinated mice

To evaluate the safety of oral vaccine using genetically modified *L.lactis*, BALB/c mice were first fed with triple vaccine once a day 5 times per week for 4 weeks (20 doses) and then sensitized and challenged with recombinant Per a 2/Der p 2/Cla c14 allergens. No major change in feed consumption was observed among any of the groups of mice. Final body weight and relative organ weights of all mice were measured after sacrifice (Table 3). There were no significant differences in body weight and organ weights among groups, indicating that there were no vaccine-specific changes. Moreover, histopathological examination of hematoxylin and eosin (H&E)-stained liver and kidney showed normal hepatic architecture with central vein and surrounding hepatocytes, and normal glomeruli and renal tubules in the kidneys of mice in all groups (Fig 8).

## Discussion

The increasing prevalence of allergic diseases is having an enormous impact on patients' quality of life. Management that is capable of improving chronic allergic inflammation without

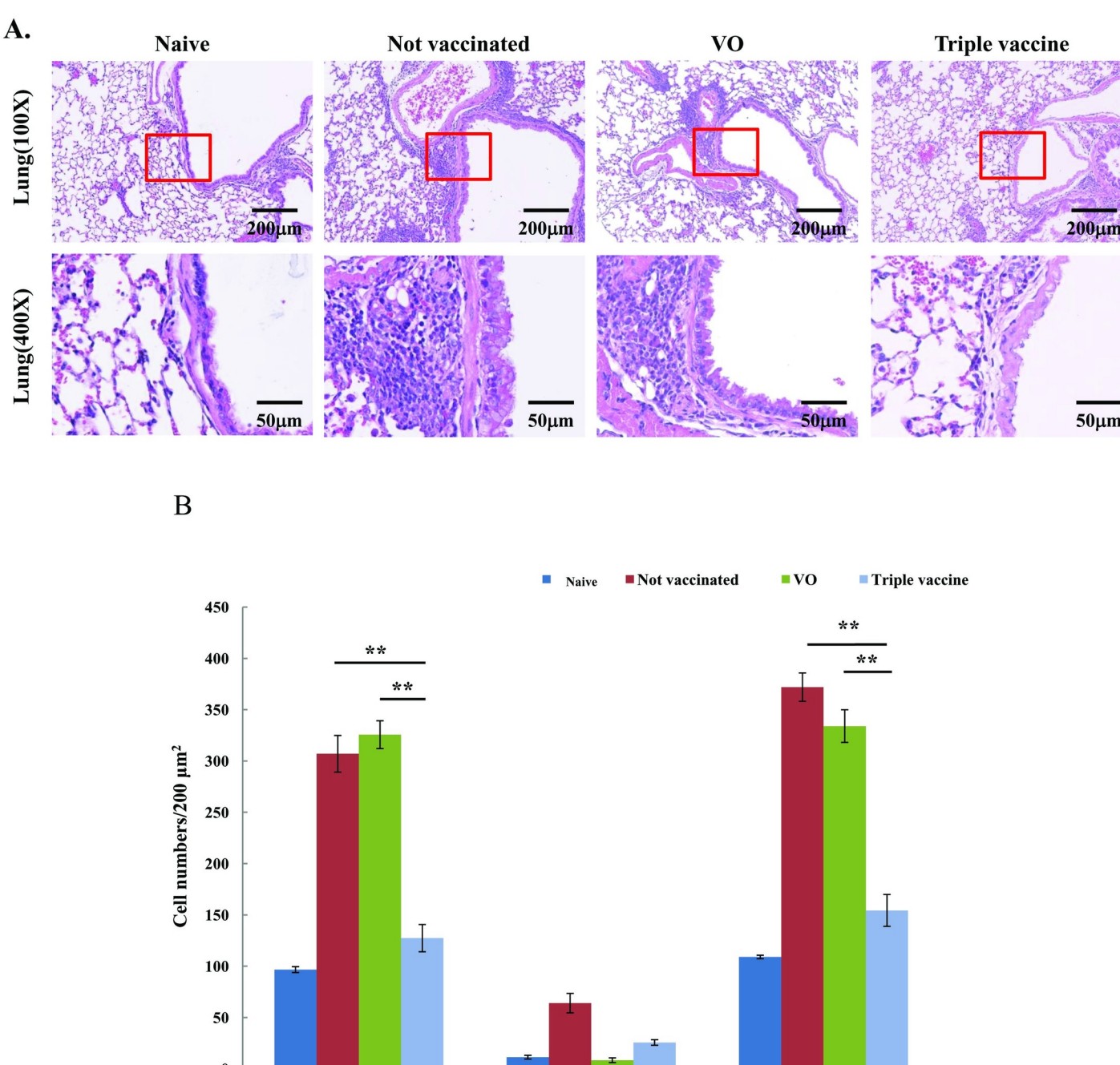

**Fig 7. The effects of triple vaccine on histopathology of lung.** (A) The representative lung sections obtained 24 hours after intratracheal challenge and stained with H&E. (B) The infiltrating inflammatory cells were quantified by light microscope under 400-fold view. The results are expressed as mean±SD of the cell numbers. The statistical significance of differences among groups was assessed by the Bonferroni test. **denoted $p < 0.01$.

impairing patients' immune function is especially important during the current COVID-19 pandemic [30, 31]. It has been reported that antecedents of allergic diseases might influence hospitalization risk in relatively young patients [32]. A recent Korean nationwide cohort showed that patients with allergic rhinitis and asthma have a greater risk of SARS-CoV-2 infection as well as worse clinical outcomes of COVID-19 [33]. The COVID-19 pandemic has also

**Table 3. Body and organ weights of the mice.**

| Groups | Body weight (g) | Liver (g) | Spleen (g) | Kidney (g) |
|---|---|---|---|---|
| Naïve | 24.27±1.07 | 1.23±0.06 | 0.12±0.01 | 0.39±0.02 |
| Not vaccinated | 23.23±1.54 | 1.21±0.12 | 0.14±0.03 | 0.35±0.02 |
| VO | 22.34±0.92 | 1.17±0.05 | 0.14±0.03 | 0.33±0.03 |
| Triple vaccine | 23.7±1.94 | 1.21±0.19 | 0.16±0.03 | 0.37±0.04 |

Data are expressed as mean±SD for each group.

changed clinical practice in the treatment of allergy, as the number of conventional injectable allergen-specific immunotherapies has decreased markedly [34]. The COVID-19 pandemic has revealed the unmet need for a safe, effective, and convenient allergen-specific immunotherapy vaccine that can be administered at home.

In this study, we established an *in vivo* mouse model to investigate whether a combination of the three major perennial indoor aeroallergens, engineered in *L. lactis* as a live vaccine, can prevent common airway allergy. Our data show that a total of only 20 doses of oral *L.lactis* triple vaccine resulted in a significant decrease in clinical bronchial hyper-responsiveness and inflammatory cell infiltration in the lungs of the vaccinated mice compared with the control mice. The protective effect was associated with an upregulation of specific IgG2a, one of the surrogate markers of successful allergen-specific immunotherapy in mice, and decreases in IL-4 and IL-13 expressions in the lung, which may affect specific IgE response. The results open the window to the possibility of developing a new convenient route of oral allergen-specific immunotherapy that could be safely administered by the patient at home. A similar idea was demonstrated by Saito et al. They used a transgenic rice seed-based oral allergy vaccine to treat Japanese cedar pollen allergic Japanese monkeys, and noted a suppression of allergen-specific cell proliferation after only two months of administration [35].

Many Lactic acid bacteria (LAB) are generally regarded as safe by the American Food and Drug Administration. *L. lactis* is the model microorganism in LAB research as it is a potential carrier of heterologous genes. Our study also demonstrated the safety of this oral vaccine as

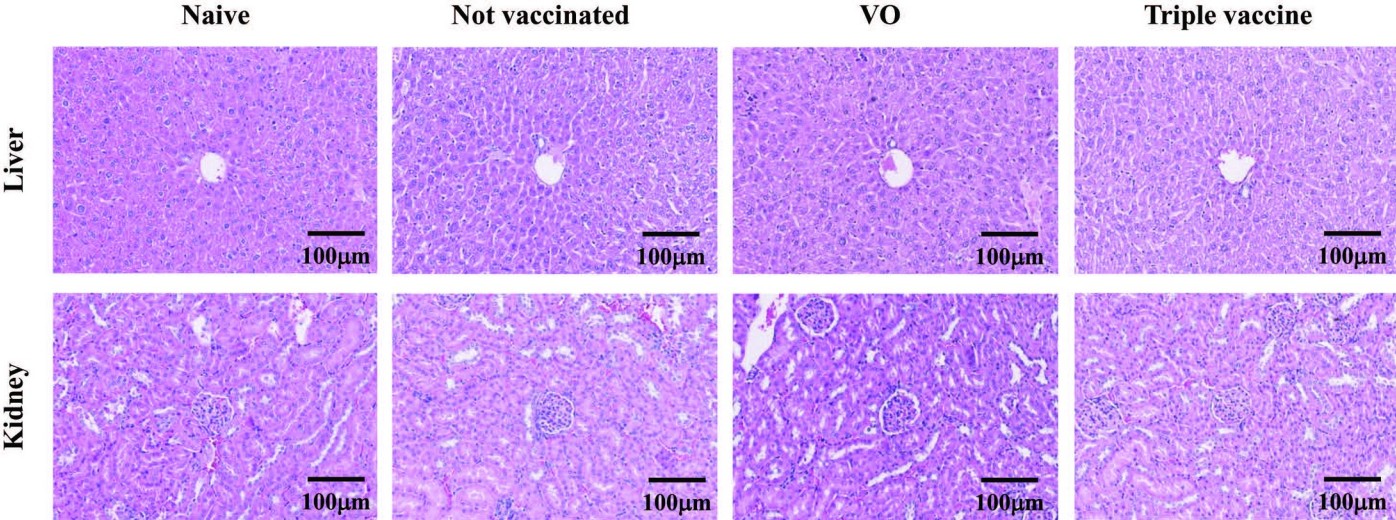

**Fig 8. Hematoxylin and eosin (H&E)-stained liver and kidney sections of mice from four groups.** The results showed normal hepatic architecture with central vein and surrounding hepatocytes, and normal glomeruli and renal tubules in the kidneys from all groups of mice. Magnification × 200.

none of the mice demonstrated signs of illness/loss of appetite during administration and there were no changes in body weight and organ weight in any of the groups.

There were some limitations in this study. As we did not check inflammatory cells, cytokines and secretary IgE, IgG isotypes and IgA in the nasal secretion and bronchial lavage fluid, whether this triple vaccine is able to induce both local and systemic tolerance required further studies. As a proof-of-concept triple oral allergen vaccine, the scope of this investigation was restricted to the prophylactic effects of the oral vaccine. Further investigation is required to establish whether or not this oral vaccine is effective in established allergic subjects, as in the study by Saito et al. Also, it is unclear whether the oral vaccine has a dose-dependent response and thus further study is needed. However, this pilot study opens the possibility that this type of therapy might be applied in tailor-made component-resolving immunotherapy depending on the individual allergen profile of each patient in the future.

In conclusion, this is the first report to describe the development of a triple-aeroallergen vaccine using a food-grade lactococcal expression system. We developed a convenient oral delivery system using a mouse model and intend to extend these results with the goal of developing a vaccination that can be self-administered at home by patients.

## Supporting information

**S1 Raw images.**
(ZIP)

## Author Contributions

**Conceptualization:** Mey-Fann Lee, Yi-Hsing Chen.

**Data curation:** Shyh-Jye Lin, Chi-Sheng Wu.

**Formal analysis:** Mey-Fann Lee, Chi-Sheng Wu.

**Investigation:** Mey-Fann Lee, Chu-Hui Chiang.

**Methodology:** Mey-Fann Lee, Chu-Hui Chiang, Chi-Sheng Wu.

**Project administration:** Yi-Hsing Chen.

**Supervision:** Yi-Hsing Chen.

**Validation:** Shyh-Jye Lin.

**Writing – original draft:** Mey-Fann Lee, Yi-Hsing Chen.

**Writing – review & editing:** Yi-Hsing Chen.

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
