## [Decision Letter · Decision Letter 0]

3 Aug 2021

PONE-D-21-11729

Effectiveness and Safety of oral lactococci-based vaccine encoding triple common allergens to prevent airway allergy in mice

PLOS ONE

Dear Dr. Chen,

Thank you for submitting your manuscript to PLOS ONE. After careful consideration, we feel that it has merit but does not fully meet PLOS ONE’s publication criteria as it currently stands. Therefore, we invite you to submit a revised version of the manuscript that addresses the points raised during the review process.

Regarding the experiments, the authors should complete the data by including the evaluation of antibodies (IgE, IgA and IgG2a) in samples collected from the airways (lung supernatants or BAL, for instance).

The authors should also comment on the interval chosen for analyzes of the adverse effects measured by histological evaluation of organs as well as other parameters that could be applied.

The final text (introduction, material and methods, discussion) should be improved according to the considerations of the reviewers.

We look forward to receiving your revised manuscript.

Kind regards,

Maria Leonor S Oliveira, PhD

Academic Editor

PLOS ONE

Journal Requirements:

2. As part of your revision, please complete and submit a copy of the ARRIVE Guidelines checklist, a document that aims to improve experimental reporting and reproducibility of animal studies for purposes of post-publication data analysis and reproducibility: https://www.nc3rs.org.uk/arrive-guidelines. Please include your completed checklist as a Supporting Information file. Note that if your paper is accepted for publication, this checklist will be published as part of your article.

3. To comply with PLOS ONE submissions requirements, in your Methods section, please provide additional information on the animal research and ensure you have included details on (1) methods of sacrifice, (2) methods of anesthesia and/or analgesia, and (3) efforts to alleviate suffering.

6. Please amend the manuscript submission data (via Edit Submission) to include author Chi-Sheng Wu.

7. Please upload a new copy of Figures 1 and 3 as the detail is not clear. Please follow the link for more information: https://blogs.plos.org/plos/2019/06/looking-good-tips-for-creating-your-plos-figures-graphics/" https://blogs.plos.org/plos/2019/06/looking-good-tips-for-creating-your-plos-figures-graphics/.

Reviewers' comments:

Reviewer's Responses to Questions

**Comments to the Author**

1. Is the manuscript technically sound, and do the data support the conclusions?

Reviewer #1: Yes

Reviewer #2: Yes

2. Has the statistical analysis been performed appropriately and rigorously? 

Reviewer #1: N/A

Reviewer #2: Yes

3. Have the authors made all data underlying the findings in their manuscript fully available?

Reviewer #1: Yes

Reviewer #2: Yes

4. Is the manuscript presented in an intelligible fashion and written in standard English?

Reviewer #1: No

Reviewer #2: Yes

5. Review Comments to the Author

Reviewer #1: I find the manuscript interesting, considering the approach of a potential preventive strategy to avoid allergies to certain allergens. The studies carried out both to obtain recombinant L. lactis for 3 antigens and the assays in a mice experimental model involve laborious work. However, I have some suggestions and inquiries to the authors:

1- I suggest to improve the introduction, to avoid the repetition of concepts (lines from 50 to 59)

2- L. lactis is simple to cultivate and it has the ability to secrete recombinant proteins to the growth medium, which together facilitates downstream processing of recombinant protein. In this work the proteins are not secreted, are produced in cytoplasm. If this is so, it should be clearly redacted in Material and Methods.

3- The authors evaluated the potential adverse effects of vaccination outside the lung through very general parameters such as weight of mice, weight of spleen, liver and kidney. It is likely that the animals will need to be monitored longer to see changes in these parameters. Histological studies were carried out in kidney and liver and they are important. However, there are simple serological parameters that provide specific data on kidney (creatinine levels in serum) and liver functionality (AST and ALT enzyme blood levels), for example. It would have been interesting to evaluate these parameters.

4- The IgE and IgG2a levels were evaluated in serum. I consider it is important to evaluate these parameters in bronchoalveolar lavage (BAL).

5- Methacholine is an inhaled drug that causes a slight narrowing of the airways in the lungs. The methacholine challenge test is a type of bronchial challenge test used to help diagnose asthma. I think that to include a brief sentence about this would facilitate the understanding of non-expert researchers on the subject.

5- In Table 2: Il should be changed by IL.

6- In Figures 4 (AB, CD, EF) Is it possible used the same scale for IgE and IgG2a?

7- The authors mentioned the limitations of this study but considering globally the results, I think the discussion could be improved.

Reviewer #2: The manuscript is well-written and describes, in a concise way, the aims of this investigation. Vaccine-based interventions represent a cost-effective alternative to prevent and treat many diseases, especially those related to hypersensitivities. Although this formulation is not the pioneer in the prevention or treatment of allergies, the research's originality relies on a combined allergens (expressed in GRAS) strategy. In this context, the work is relevant and consists of an original contribution to a preventive as well as potential therapeutic use of vaccines. The manuscript also reveals a substantial amount of work and the use of adequate techniques to characterize their system.

There are just some minor questions that would be addressed by the authors:

1. In Materials and Methods (lines 169-180), the authors describe the immunization protocol, with specific doses, besides the sensitization and challenge methods. It is not clear how they made this experimental design, since this section does not bring any references. These protocols, mentioned previously, were not referenced. Therefore, I recommend that the authors add more information about the experimental design regarding these methods.

2. The information of protein sizes (in kDa) is lacking in the text (Materials and Methods and Results as well). I suggest that this piece of information appears in Results section (for instance, between lines 246-254).

3. The statistical analysis was not properly described in the text (lines 240-243). The authors only mention that "appropriate methods" were performed to analyze the results. The specific information about statistics of antibodies levels can be found only in the figure 4 description, for instance.

4. I was wondering why the authors did not collect nasal as well as bronchoalveolar lavages to determine IgA levels. Hypersensitivities that affect respiratory tracts (upper and lower) usually result in local and systemic response, characterized by IgG (from Th2 profile), IgE and secretory IgA production. For this reason, the authors would address this question, in other words, they should clarify the reason why this isotype was not analyzed.

If the authors address these questions, the manuscript will be suitable for publication at PLoS One.

6. PLOS authors have the option to publish the peer review history of their article (what does this mean?). If published, this will include your full peer review and any attached files.

Reviewer #1: No

Reviewer #2: No

---

## [Author Response · Author response to Decision Letter 0]

31 Oct 2021

Dear Dr. Oliveira and the reviewers:

Thank you so much for reviewing our manuscript. Please find below our point-by-point responses to the reviewers’ comments. We gratefully appreciate the valuable suggestions. 

Reviewer #1: I find the manuscript interesting, considering the approach of a potential preventive strategy to avoid allergies to certain allergens. The studies carried out both to obtain recombinant L. lactis for 3 antigens and the assays in a mice experimental model involve laborious work. However, I have some suggestions and inquiries to the authors:

Question 1. I suggest to improve the introduction, to avoid the repetition of concepts (lines from 50 to 59)

Reply: We have condensed the last part of introduction (original lines 50-59) to 5 lines (new lines 51-55) as the recommendation of the reviewer. 

Question 2. L. lactis is simple to cultivate and it has the ability to secrete recombinant proteins to the growth medium, which together facilitates downstream processing of recombinant protein. In this work the proteins are not secreted, are produced in cytoplasm. If this is so, it should be clearly redacted in Material and Methods.

Reply: Thank you for the reminding. The description has been added to the revised manuscript. Please see lines 90-91: ” In this study, the recombinant allergens Per a 2, Der p 2, and Cla c 14 were intracellularly produced under nisin induction in L. lactis NZ3900 clones.”

Question 3. The authors evaluated the potential adverse effects of vaccination outside the lung through very general parameters such as weight of mice, weight of spleen, liver and kidney. It is likely that the animals will need to be monitored longer to see changes in these parameters. Histological studies were carried out in kidney and liver and they are important. However, there are simple serological parameters that provide specific data on kidney (creatinine levels in serum) and liver functionality (AST and ALT enzyme blood levels), for example. It would have been interesting to evaluate these parameters.

Reply: Thank you for the comments. We love to have more biochemical safety data if possible. Unfortunately, after doing the immunological testing, the residual amount of the blood drawn from retro-orbital space of each tiny mouse was not enough to check these biochemical tests. However, we have data from rabbits to look at the safety of this triple-allergen vaccine using a 13-week oral toxicity study (data see below table). The routine hematological and serum biochemical parameters were monitored monthly and the data are summarized in the following tables. The serum liver protein, including aspartate transaminase (AST), alanine transaminase (ALT), albumin (ALB) and total protein (TP); renal function indicators: blood urea nitrogen (BUN) and creatinine (CRE); and serum cholesterol (CHOL) and glucose (GLU) levels were not significantly different between the groups receiving oral administration of Triple vaccine and the control group (NC) in either male or female animals, and all data were within the normal ranges.

Question 4. The IgE and IgG2a levels were evaluated in serum. I consider it is important to evaluate these parameters in bronchoalveolar lavage (BAL).

Reply: Thank you for the comments. IgE and IgE-allergen immune complex in the airway are indeed good markers for airway inflammation. It is unfortunate we did not look at IgE and IgG2a in BAL fluid in this study. The statement has been added to the study limitation in the Discussion section lines 341-343. 

Question 5. Methacholine is an inhaled drug that causes a slight narrowing of the airways in the lungs. The methacholine challenge test is a type of bronchial challenge test used to help diagnose asthma. I think that to include a brief sentence about this would facilitate the understanding of non-expert researchers on the subject.

Reply: The description regarding methacholine challenge was added in the Section of Materials and Methods lines 183-184.” Methacholine is an inhaled drug that causes narrowing of the airways in the lungs. The methacholine challenge test is a type of bronchial challenge test used to help diagnose asthma.”

Question 5. In Table 2: Il should be changed by IL.

Reply: The typo has been corrected in the new Table 2 as recommended by the reviewer.

Question 6. In Figures 4 (AB, CD, EF) Is it possible used the same scale for IgE and IgG2a?

Reply: A new Figure 4 with the Y-axis on the same scale has been made as recommended by the reviewer. 

Question 7. The authors mentioned the limitations of this study but considering globally the results, I think the discussion could be improved.

Reply: We have added more discussion regarding the study limitations as the comments of the reviewer.

Reviewer #2: The manuscript is well-written and describes, in a concise way, the aims of this investigation. Vaccine-based interventions represent a cost-effective alternative to prevent and treat many diseases, especially those related to hypersensitivities. Although this formulation is not the pioneer in the prevention or treatment of allergies, the research's originality relies on combined allergens (expressed in GRAS) strategy. In this context, the work is relevant and consists of an original contribution to a preventive as well as potential therapeutic use of vaccines. The manuscript also reveals a substantial amount of work and the use of adequate techniques to characterize their system.

There are just some minor questions that would be addressed by the authors:

Question 1. In Materials and Methods (lines 169-180), the authors describe the immunization protocol, with specific doses, besides the sensitization and challenge methods. It is not clear how they made this experimental design, since this section does not bring any references. These protocols, mentioned previously, were not referenced. Therefore, I recommend that the authors add more information about the experimental design regarding these methods.

Reply: We have specifically cite a reference describing our previous study as recommended by the reviewer. Please see lines 143-144: The mouse model was established as previously described[29] .

Question 2. The information of protein sizes (in kDa) is lacking in the text (Materials and Methods and Results as well). I suggest that this piece of information appears in Results section (for instance, between lines 246-254).

Reply: More details regarding the sizes of cDNA and the corresponding recombinant proteins were added in new Table 1. Please see line 73-74: The amplified sizes of cDNA and the molecular weights of the derived proteins are also shown in new Table 1.

Question 3. The statistical analysis was not properly described in the text (lines 240-243). The authors only mention that "appropriate methods" were performed to analyze the results. The specific information about statistics of antibodies levels can be found only in the figure 4 description, for instance.

Reply: Yes. The detailed information regarding statistic methods has been added to the revised manuscript. Please see lines 217-218: All values are expressed as means ± SD, and differences between groups were analyzed by one-way analysis of variance with a Bonferroni multiple comparison test.

Question 4. I was wondering why the authors did not collect nasal as well as bronchoalveolar lavages to determine IgA levels. Hypersensitivities that affect respiratory tracts (upper and lower) usually result in local and systemic response, characterized by IgG (from Th2 profile), IgE and secretory IgA production. For this reason, the authors would address this question, in other words, they should clarify the reason why this isotype was not analyzed.

Reply: Thank you for the comments. We have added sentences regrading study limitation in the Discussion section lines 341-343 “As we did not check inflammatory cells, cytokines and secretary IgE, IgG isotypes and IgA in the nasal secretion and bronchial lavage fluid, whether this triple vaccine is able to induce both local and systemic tolerance required further studies.”.

---

## [Decision Letter · Decision Letter 1]

1 Dec 2021

Effectiveness and Safety of oral lactococci-based vaccine encoding triple common allergens to prevent airway allergy in mice

PONE-D-21-11729R1

Dear Dr. Chen,

We’re pleased to inform you that your manuscript has been judged scientifically suitable for publication and will be formally accepted for publication once it meets all outstanding technical requirements.

Kind regards,

Maria Leonor S Oliveira, PhD

Academic Editor

PLOS ONE

Additional Editor Comments (optional):

Reviewers' comments:

Reviewer's Responses to Questions

**Comments to the Author**

1. If the authors have adequately addressed your comments raised in a previous round of review and you feel that this manuscript is now acceptable for publication, you may indicate that here to bypass the “Comments to the Author” section, enter your conflict of interest statement in the “Confidential to Editor” section, and submit your "Accept" recommendation.

Reviewer #1: All comments have been addressed

Reviewer #2: All comments have been addressed

2. Is the manuscript technically sound, and do the data support the conclusions?

Reviewer #1: Yes

Reviewer #2: Yes

3. Has the statistical analysis been performed appropriately and rigorously? 

Reviewer #1: Yes

Reviewer #2: Yes

4. Have the authors made all data underlying the findings in their manuscript fully available?

Reviewer #1: Yes

Reviewer #2: Yes

5. Is the manuscript presented in an intelligible fashion and written in standard English?

Reviewer #1: Yes

Reviewer #2: Yes

6. Review Comments to the Author

Reviewer #1: The manuscript was substantially improved.

I think that the authors responded favorably to the suggestions and questions raised, and in its current form, the manuscript should be accepted for publication.

Reviewer #2: The authors presented a document with all comments addressed. I would like to add some minor correction regarding the answer to question 4 of my comments. It is a matter of misspelling. They must write "secretory" instead of "secretary" and "requires" instead of "required". In spite of this minor issue, the paper is ready for publication.

7. PLOS authors have the option to publish the peer review history of their article (what does this mean?). If published, this will include your full peer review and any attached files.

Reviewer #1: No

Reviewer #2: No

---

## [Editor Report · Acceptance letter]

9 Dec 2021

PONE-D-21-11729R1 

Effectiveness and Safety of oral lactococci-based vaccine encoding triple common allergens to prevent airway allergy in mice 

Dear Dr. Chen:

I'm pleased to inform you that your manuscript has been deemed suitable for publication in PLOS ONE. Congratulations! Your manuscript is now with our production department. 

Kind regards, 

on behalf of

Dr. Maria Leonor S Oliveira 

Academic Editor

PLOS ONE